

# Deep learning prediction of mild cognitive impairment conversion to Alzheimer's disease at 3 years after diagnosis using longitudinal and whole-brain 3D MRI

Ethan Ocasio and Tim Q. Duong

Department of Radiology, Montefiore Medical Center, Albert Einstein College of Medicine, Bronx, NY, United States of America

Corresponding author
Tim Q. Duong,
tim.duong@einsteinmed.org

## ABSTRACT

**Background**. While there is no cure for Alzheimer's disease (AD), early diagnosis and accurate prognosis of AD may enable or encourage lifestyle changes, neurocognitive enrichment, and interventions to slow the rate of cognitive decline. The goal of our study was to develop and evaluate a novel deep learning algorithm to predict mild cognitive impairment (MCI) to AD conversion at three years after diagnosis using longitudinal and whole-brain 3D MRI.

**Methods**. This retrospective study consisted of 320 normal cognition (NC), 554 MCI, and 237 AD patients. Longitudinal data include T1-weighted 3D MRI obtained at initial presentation with diagnosis of MCI and at 12-month follow up. Whole-brain 3D MRI volumes were used without a priori segmentation of regional structural volumes or cortical thicknesses. MRIs of the AD and NC cohort were used to train a deep learning classification model to obtain weights to be applied via transfer learning for prediction of MCI patient conversion to AD at three years post-diagnosis. Two (zero-shot and fine tuning) transfer learning methods were evaluated. Three different convolutional neural network (CNN) architectures (sequential, residual bottleneck, and wide residual) were compared. Data were split into 75% and 25% for training and testing, respectively, with 4-fold cross validation. Prediction accuracy was evaluated using balanced accuracy. Heatmaps were generated.

**Results**. The sequential convolutional approach yielded slightly better performance than the residual-based architecture, the zero-shot transfer learning approach yielded better performance than fine tuning, and CNN using longitudinal data performed better than CNN using a single timepoint MRI in predicting MCI conversion to AD. The best CNN model for predicting MCI conversion to AD at three years after diagnosis yielded a balanced accuracy of 0.793. Heatmaps of the prediction model showed regions most relevant to the network including the lateral ventricles, periventricular white matter and cortical gray matter.

**Conclusions**. This is the first convolutional neural network model using longitudinal and whole-brain 3D MRIs without extracting regional brain volumes or cortical thicknesses to predict future MCI to AD conversion at 3 years after diagnosis. This approach could lead to early prediction of patients who are likely to progress to AD and thus may lead to better management of the disease.

## BACKGROUD

Alzheimer's disease (AD) is a progressive neurodegenerative disease characterized by loss of memory and other cognitive functions (*McKhann et al., 2011*). Mild cognitive impairment (MCI) is considered a transitional state between normal aging and dementia. Many patients progress from MCI to AD, but others remain stable without developing AD. Although diagnoses of MCI and AD are typically made using neuropsychological tests (*Petersen et al., 1999*; *Jak et al., 2009*), imaging methods are also used to diagnose AD and to monitor disease progression because they provide neural correlates of the underlying brain dysfunction in a longitudinal non-invasive manner (*Johnson et al., 2012*). While there is no cure for AD, early diagnosis and accurate prognosis may enable or encourage lifestyle changes, neurocognitive enrichment, and therapeutic interventions that strive to improve symptoms, or at least slow down mental deterioration, thereby improving the quality of life (*Epperly, Dunay & Boice, 2017*).

Machine learning (ML) is increasingly being used in medicine from disease classification to prediction of clinical progression (*De Bruijne, 2016*; *Erickson et al., 2017*). ML uses algorithms to learn the relationship amongst different data elements to inform outcomes. Neural networks, a form of ML, are made up of a collection of connected nodes that model the neurons present in a human brain (*Graupe, 2013*). Each connection, similar to a synapse, transmits and receives signals to other nodes. Each node and the connections it forms are initialized with weights which are adjusted throughout training and create mathematical relationships between the input data and the outcomes. In contrast to traditional analysis methods such as logistic regression, neural networks do not require relationships between different input variables and the outcomes to be explicitly specified a priori. In radiology, ML can accurately detect lung nodules on chest X-rays (*Harris et al., 2019*). In cardiology, ML can detect abnormal EKG patterns (*Johnson et al., 2018*). ML has also been used to estimate risk, such as in the Framingham Risk Score for coronary heart disease (*Alaa et al., 2019*), and to guide antithrombotic therapy in atrial fibrillation (*Lip et al., 2010*) and defibrillator implantation in hypertrophic cardiomyopathy (*O'Mahony et al., 2014*). Convolutional neural networks (CNNs), a deep-learning method, are widely used for image analysis and analysis of complex data (*Lecun et al., 1998*; *Krizhevsky, Sutskever & Hinton, 2012*; *Simonyan & Zisserman, 2014*).

Deep learning classification amongst normal cognition (NC), MCI and AD based on magnetic resonance imaging (MRI) data have been reported (*Cheng et al., 2017*; *Korolev et al., 2017*; *Wen et al., 2020*). By contrast, there are comparatively fewer studies that reported prediction of MCI to AD conversion using deep learning of MRI data (*Lian et al., 2018*; *Lin et al., 2018*; *Liu et al., 2018*; *Shmulev & Belyaev, 2018*; *Basaia et al., 2019*; *Wen et al., 2020*). A few ML studies used extracted brain structures or cortical thicknesses, and some used 3D patches from predetermined locations across the brain, but not whole-brain MRI data, to

predict MCI to AD conversion (*Lian et al., 2018*; *Liu et al., 2018*; *Wen et al., 2020*). Most of the few prediction studies used single timepoint MRI data. To our knowledge there are only two studies that predicted disease progression using longitudinal imaging data. Bhagwat used a neural network (albeit not deep learning) and extracted cortical thicknesses from MRIs at two time points to predict decline in Mini-Mental Status Exam (MMSE) scores (*Bhagwat et al., 2018*). Ostertag et al. used a CNN model on whole-brain MRI at two time points to predict decline in MMSE score but did not test their model on an independent testing dataset (*Ostertag, Beurton-Aimar & Urruty, 2019*). These two studies mixed NC, MCI and AD participants and thus accuracies are not applicable to prediction of MCI to AD conversion. *To our knowledge, there are no published studies to date on deep learning to predict MCI to AD conversion using longitudinal and whole-brain 3D MRI.*

The goal of our study was thus to develop and evaluate a novel deep-learning algorithm to predict MCI to AD conversion at three years after diagnosis using longitudinal and whole-brain 3D MRI. Longitudinal data include MRI obtained at initial presentation with diagnosis of MCI and at 12-month follow up. Whole-brain 3D MRI volumes were used without a priori segmentation of regional structural volumes or cortical thicknesses. Several convolutional model architectures, transfer learning methods, and methods of merging longitudinal whole-brain 3D MRI data were evaluated to derive the final optimal deep-learning predictive model.

## METHODS

Figure 1 shows the overall design of the experiment. 3D MRIs of the AD and NC cohort were trained in a CNN classification model to obtain weights for transfer learning to be used in the CNN prediction of MCI patient conversion to AD in 3 years after diagnosis. Two (zero-shot and fine tuning) transfer learning methods for prediction were evaluated (*Pan & Yang, 2010*). The zero-shot transfer method used the intact weights obtained from the NC-AD classification without any additional training. The fine-tuning transfer method kept the weights in the convolutional layers frozen while allowing the remaining fully connected layers to change during additional training against the MCI images.

### Participants

Data used in this study was obtained from the Alzheimer's Disease Neuroimaging Initiative (ADNI) database (adni.loni.usc.edu). Patients were taken from the ADNI1, ADNIGO, ADNI2, and ADNI3 patient sets. For the prediction task, inclusion criteria were patients diagnosed with MCI at baseline with MRI taken at baseline and ~12 months after baseline, and with a final diagnosis at 3 years post baseline of either MCI (labeled as stable MCI or sMCI) or AD (labeled as progressive MCI or pMCI). Patients who converted from MCI to AD before their follow up image were excluded since the analysis of their longitudinal image at that point would represent a diagnostic classification not a prediction. Table 1 summarizes the participant demographics. Data were split into 75% and 25% for training/validation and testing, respectively, with the training/validation set composed of 415 patients (249 sMCI and 166 pMCI), and the testing set composed of 139 patients (84 sMCI and 55 pMCI). Then, we optimized the networks using a 4-fold cross-validation

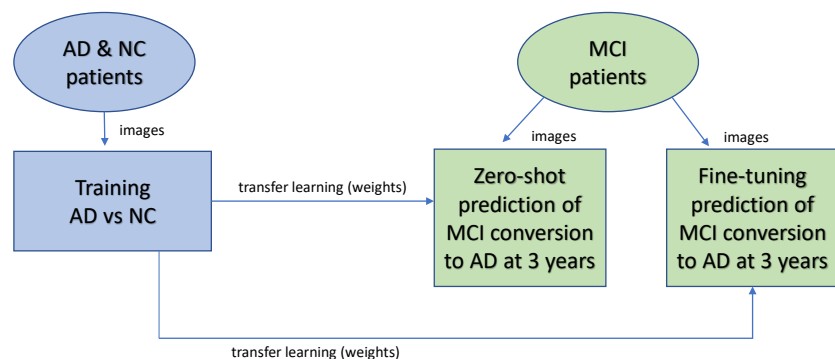

**Figure 1** **Overview of experimental design.** AD and NC MRI data were first trained to obtain weights (a classification task) for transfer learning (blue). After training, the weights are transferred to the prediction task (green) to predict whether patients will remain stable or progress within three years. Two different transfer learning methods were studied. With zero-shot, no further training was performed after the transfer, so the MCI images were analyzed for prediction by the network with the same weights copied over from the classification task. With fine-tuning, after weights are copied over from the classification task for initialization, additional training is performed against the MCI image data.

**Table 1** **Summary of participant demographics.**

|  | N | Age (years) | Gender |
| --- | --- | --- | --- |
| NC | 320 | $75.7 \pm 5.6$ [60.9, 90.8] | 160 F / 160 M |
| AD | 237 | $75.9 \pm 7.9$ [56.1, 92.0] | 108 F / 129 M |
| sMCI | 333 | $72.7 \pm 7.6$ [56.1, 89.8] | 137 F / 196 M |
| pMCI | 221 | $75.2 \pm 7.2$ [55.6, 89.5] | 92 F / 129 M |

**Notes.**

NC, normal cognition; AD, Alzheimer's disease; sMCI, stable mild cognitive impairment; pMCI, progressive mild cognitive impairment; F, Female; M, Male.

Age values are mean $\pm$ standard deviation [range]

on the training/validation set, resulting in 56.25% for training and 18.75% for validation from the complete data in each fold split. In the end, we had four trained models for each experiment configuration, and reported the mean and standard deviation (SD) BA.

For the AD-NC classification to obtain weights for transfer learning, a separate group of participants with a baseline diagnosis of AD or NC with images taken at baseline and ~12 months from baseline were selected. Data for training/validation and testing were randomly split up as 70% and 30% respectively. The training/validation set consisted of 387 patients (160 AD and 227 NC), while the independent testing set consisted of 170 patients (77 AD and 93 NC). Using 4-fold cross validation on the training/validation set resulted in 52.5% for training and 17.5% for validation from the complete data in each fold split. The networks were trained to classify the patients between AD and NC against the ground truth diagnosis of each patient using ADNI criteria. For all experiments the study size represents the total available patients in ADNI database fully meeting all inclusion criteria.

## Preprocessing

3D volumes of T1-weighted MRI were used as input to the networks. To remove intensity inhomogeneity from the image inputs, T1-weighted images with nonparametric non-uniform intensity normalization (N3) correction (*Sled, Zijdenbos & Evans, 1998*) were selected from the ADNI database. All MRIs were skull stripped with DeepBrain (*Itzcovich, 2018*), then linearly registered against a 2 mm standard brain with nine degrees of freedom (translation, rotation, scaling) using FSL FLIRT (*Jenkinson et al., 2012*), and finally min/max intensity normalized. Resulting images had a resolution of $91 \times 109 \times 91$ voxels. During training, data augmentation was performed on the training set by rotating each MRI by up to 5% in any direction and randomly flipping them left to right along the sagittal axis.

## Training, validation, and testing

Images were split into testing, validation, and testing sets at the patient level in order to avoid data leakage. For both classification and prediction tasks, after assigning labels (either AD vs NC or sMCI vs pMCI), SciPy's train_test_split function was used to randomly split training/validation and testing and cross validation (*Virtanen et al., 2020*). Stratification was done based on the labels in order to maintain a consistent distribution of diagnoses across the training, validation, and testing datasets. Randomization seed was set up as a constant to ensure the same train/validation/test split was obtained for each experiment run. Balanced accuracy (BA), defined as the average of sensitivity and specificity, was used as the main binary classifier metric to eliminate the inflated accuracy effect caused by imbalanced data sets. For purposes of computing accuracy, we use a standard value of 0.5 as the threshold between the two labels (either NC & AD or pMCI & sMCI). We also computed the area under the receiver operating characteristic curve (AUC) for each run, since it provides a more general measure of the potential performance of a network across a range of thresholds. To prevent data leakage, once the test partition was randomly selected, the test set images were set aside and not used for any training or validation. We repeated each training experiment 4 times, using the standard k-fold cross validation approach where the training/validation set is partitioned into four subsets, using each subset once for validation, and reported both the mean and standard deviation of the BAs for each experiment. This ensures that each image is included at least once in both the validation and training sets, minimizing the potential selection bias that a single random data split may introduce. For each cross-validation fold, classification task training was performed for 200 epochs with early stopping based on no improvement in loss function for 80 epochs. Fine tuning after freezing all convolutional layers was performed for 100 epochs with early stopping patience of 40. The weights of the epoch ending with the lowest loss were saved and used to obtain the validation BA, and then the network was run against the test set for the test BA. Additional attempts at fine tuning by also unfreezing the last convolutional block were noted to degrade accuracy, so this approach was not considered any further.

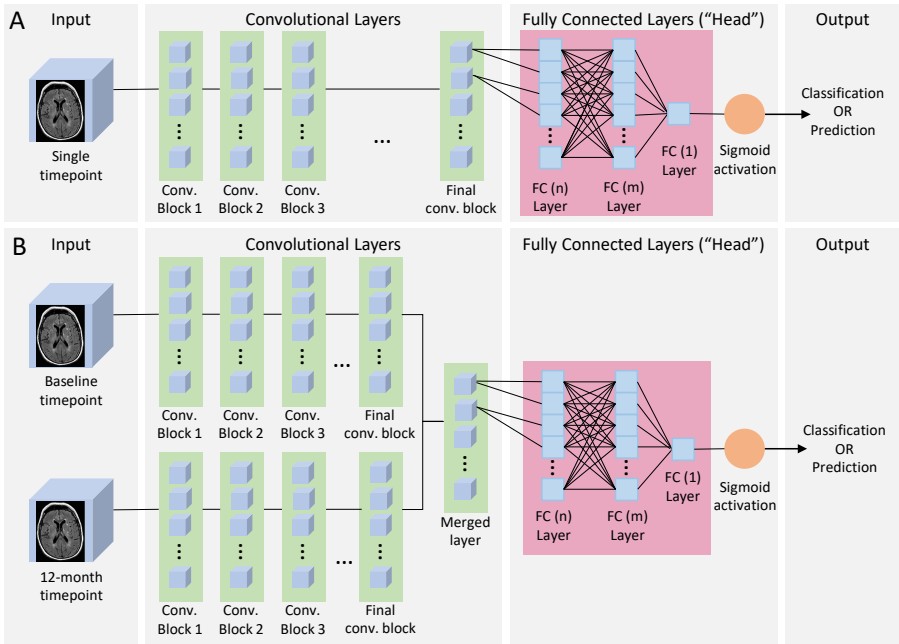

**Figure 2** **Single and dual time point CNN architecture.** (A) Single timepoint CNN. For classification, input consisted of a single timepoint full-subject 3D MRI of patients diagnosed at baseline as either AD or CN, and output was binary classification of AD vs CN. For prediction, input was a single timepoint full-subject 3D MRI of patients diagnosed as MCI and output was a binary prediction of whether the patient progressed (pMCI) or remained stable (sMCI) 3 years later. (B) Dual timepoint CNN. Input included 3D MRI images obtained at both baseline and 12 months, with the patient population and output categories identical than those used for single timepoint for classification and prediction. Both kinds of networks began with a series of convolutional blocks, followed by flattening into one or more fully connected layers ending in a final binary choice of classification or prediction.

## CNN architecture

The neural network models (Fig. 2) consisted of a convolutional section followed by a fully connected section (head). Three (sequential, residual with bottleneck, and wide residual) types of convolutional blocks (Fig. 3) and three head architectures (Fig. 4) were examined. In addition, we also evaluated these networks using a single (baseline) timepoint MRI and MRIs at two time points (baseline and 12 months). For the dual timepoint networks, three types of networks were explored to incorporate longitudinal data: (1) Siamese network (two identical parallel channels with weights tied together) using a subtraction layer as the merging function, (2) Siamese network with a concatenation merge layer, and (3) a twin network (identical channels with weights independently optimized). Since the flattened set of post-convolution features in the twin architecture is different in each channel, as they are the result of different parameters, there is no rationale for directly subtracting them, so we only considered a concatenation merge option for the twin architecture. For all networks, the final binary classifier layer was fully connected with sigmoid activation. When performing the prediction tasks (using both zero-shot learning and fine-tuning) in the single timepoint experiments, we attempted the same task using both the initial baseline MRI as well as the 1-year follow up image. Using either the single timepoint with

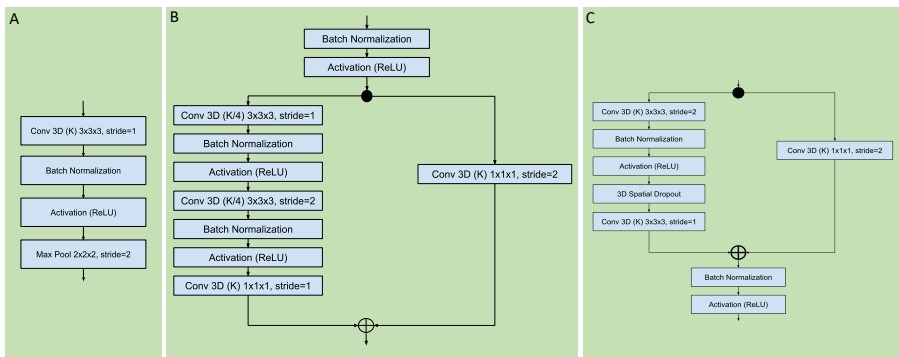

**Figure 3 Sequential, residual with bottleneck, and wide residual CNN blocks.** The convolutional layers portion of the network was organized as a series of blocks, each one with an increasing number K of activation maps (width), and with a corresponding decrease in resolution obtained by either pooling or stride during convolution. The figures detail the individual layers that compose a single block. (A) Sequential convolutional block. Each block was composed of a single $3 \times 3 \times 3$ convolution, followed by batch normalization, ReLU activation, and max pooling to reduce the resolution. (B) Residual bottleneck with pre-activation convolutional block. Convolutions were preceded by batch normalization and ReLU activation. Two bottleneck $3 \times 3 \times 3$ convolutions have a width of K/4 followed by a final $1 \times 1 \times 1$ convolution with K width. In parallel the skip residual used a $1 \times 1 \times 1$ convolution to match the width and resolution. In this architecture the first residual block was preceded by an initial batch normalization followed by a single $5 \times 5 \times 5$ convolution, plus one final batch normalization and ReLU activation after the last block (not shown). (C) Wide Residual Network convolutional block. In this architecture the batch normalization and activations occured after the convolutional layers. Each block had two $3 \times 3 \times 3$ convolutional layers with 3D spatial dropout in between, plus a $1 \times 1 \times 1$ skip residual convolution to match width and resolution.

the 1-year image or both timepoints together longitudinally represents, for patients who have had MCI for one year and not yet progressed to AD, a prediction of whether they will eventually convert to AD within two more years.

After initial experimentation, an optimal set of blocks was identified for the sequential and wide residual network styles, namely using 6 blocks with widths (number of activation maps) = {64, 128, 256, 512, 1024, 2048}. In the case of the sequential convolution network, each block reduced the resolution via maximum pooling as the width increased, down to $1 \times 1 \times 1$ in the final convolutional block. In the case of the wide residual network, convolutions with the use of strides gradually reduced the resolution down to $2 \times 2 \times 2$, with a global maximum pooling layer in the head portion of the network. When a convolutional layer processes an input whose size is odd-numbered in any of its dimensions ($2n$-1 for any integer $n$) the resulting output of a stride 2 convolution with zero-padding will be of size $n$ for the corresponding dimension. For example, since the input image has resolution $91 \times 109 \times 91$, the output after the first stride 2 convolution will be $46 \times 55 \times 46$. For the bottleneck residual, the final convolutional resolution was also $2 \times 2 \times 2$, achieved via strides, but this required 7 blocks with widths = {64, 64, 128, 256, 512, 1024, 2048}. The bottleneck architecture used $1 \times 1 \times 1$ convolutions for resolution reduction, so the behavior was slightly different when the resolution had odd numbers, allowing for 7 instead of 6 blocks until the resolution was down to $2 \times 2 \times 2$. The portion of the network with flattened non-convolutional fully connected layers after the last convolutional layer up

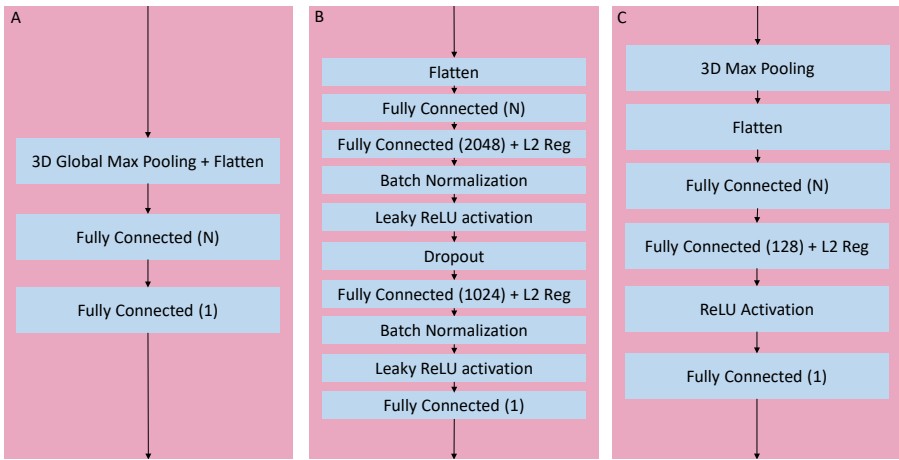

**Figure 4** **Three head architectures.** (A) 3D global maximum pooling fully connected block. The global pooling inherently flattened the nodes into a fully connected layer with N nodes directly followed by the final binary classifier layer. (B) Long fully connected block. After flattening into a layer of N nodes, there are two sets of fully connected (size 2,048 and 1,024), batch normalization, and leaky ReLU activation layers separated by a single dropout layer, before the final binary classifier. (C) Medium fully connected block. Initial 3D max pooling is followed by flattening into a fully connected layer of size N followed by an additional fully connected layer of size 128 and ReLU activation.

until the final binary classifier is known as the "head." After initial analysis of networks with varying heads, a global maximum pooling operation resulting in a fully connected layer with a number of nodes equal to the number of activation maps (width) of the last convolutional step, followed directly by a single final dense prediction layer, was selected as the optimal fully connected layer architecture (Fig. 4A).

Training was initially attempted using both non-adaptive (SGD with Nesterov momentum), as well as adaptive (Adam) optimizers (*Kingma & Ba, 2019*). The Adam optimizer was able to achieve reductions in loss function with accompanying increase in accuracy much more rapidly and aggressively. However, without a scheduled reduction of the base learning rate, the network became unstable in the latter epochs with rapid swings in the loss function. The use of an exponentially decaying learning rate schedule consistently stabilized both the loss and accuracy curves in an optimal fashion. The final selected optimization approach was thus the Adam optimizer with an exponentially decreasing learning rate schedule with expected initial rate $LR_{start}$ and final rate $LR_{end}$ where $t$ is the current epoch and $T$ is the final expected number of epochs:

$$LR_{epoch} = LR_{start} \left( \frac{LR_{end}}{LR_{start}} \right)^{t/T}$$

L2 regularization as also added in the convolutional layers in all networks. For the sequential model, we used a regularization parameter of 0.005, and for the residual models we used a parameter of 0.0001. All training was performed using Tensorflow2/Keras python library, on Google Compute Platform virtual instances with Tesla V-100 GPU acceleration.

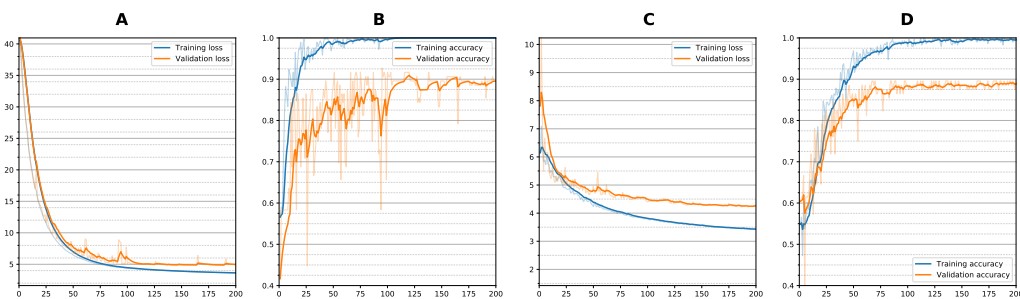

**Figure 5** **Training curves during classification.** Loss and Accuracy curves during training for both training and validation sets. For sequential network and single timepoint, (A) loss, (B) accuracy. For wide residual network and dual timepoints, (C) loss, (D) accuracy. Solid lines are smoothed with 0.8 factor and faint lines show the unsmoothed values for each epoch.

### Network visualization by heatmap

To visualize the brain regions that are most relevant to the network, the Grad-CAM (*Selvaraju et al., 2017*) technique was modified to work in 3 dimensions for generating heatmaps. Since the models reduced the resolution of the image information within the convolution blocks down to $2 \times 2 \times 2$ voxels or less, the 3D Grad-CAM technique was applied to higher convolutional layers (with resolution close to 40 voxels per axis) to obtain more useful visualization heatmaps. This approach enabled visual highlighting of the sections of the images that were most significant to the network. Heatmaps were obtained during the execution of the prediction models.

## RESULTS

### AD versus NC classification to generate weights

Figure 5 shows training curves for sequential single channel and wide residual dual channel training for the classification experiments. Overall, loss functions converged and leveled off at around 75 epochs. Other models showed similar convergence characteristics. Table 2 details the results of the classification experiments to generate weights. For the single timepoint networks, sequential architecture performed best (BA = 0.860) followed by wide residual (BA = 0.840) and bottleneck residual (BA = 0.727) on the testing data. For the dual timepoint networks, Siamese network with subtraction performed poorly overall with all architectures (BA 0 < 0.65) and that the twin non-Siamese approach with merge concatenation performed best for dual channels. The wide residual (BA = 0.887) performed best followed by sequential (BA = 0.876) and bottleneck residual (BA = 0.800). Training time for each run was approximately 60–90 min. After model was trained, classification of a patient takes two seconds or less (most of this time is loading the images from storage into memory).

### Prediction of AD conversion at 3 years after diagnosis

Figure 6 shows the training curve for one of the dual sequential transfer learning fine-tuning attempts for the prediction experiment. Additional training did not improve the accuracy

**Table 2  AD vs NC classification to generate weights.** Balanced accuracy and area under the receiver operating characteristic curve of the validation and test datasets obtained using single and dual time point networks with sequential, bottleneck residual and wide residual CNN blocks. Dual timepoint networks were twin (equal structure), non-Siamese (separate weights) and merged using concatenation.

| | Model convolution style | Validation mean ± SD | | Test mean ± SD | |
|---|---|---|---|---|---|
| | | BA | AUC | BA | AUC |
| Single | Sequential | 0.854 ± 0.027 | 0.918 ± 0.018 | **0.860 ± 0.016** | 0.922 ± 0.005 |
| | Bottleneck Residual | 0.689 ± 0.020 | 0.774 ± 0.017 | 0.727 ± 0.051 | 0.782 ± 0.052 |
| | Wide Residual | 0.835 ± 0.025 | 0.903 ± 0.027 | 0.840 ± 0.017 | 0.917 ± 0.006 |
| Dual | Sequential | 0.855 ± 0.014 | 0.938 ± 0.007 | 0.876 ± 0.010 | 0.937 ± 0.012 |
| | Bottleneck Residual | 0.772 ± 0.046 | 0.865 ± 0.037 | 0.800 ± 0.045 | 0.869 ± 0.043 |
| | Wide Residual | 0.856 ± 0.025 | 0.942 ± 0.012 | **0.887 ± 0.009** | 0.933 ± 0.003 |

**Notes.**
BA, balanced accuracy; AUC, area under the receiver operating characteristic curve.
Best test average BAs are highlighted in bold for single channel (sequential) and dual channel (wide residual)

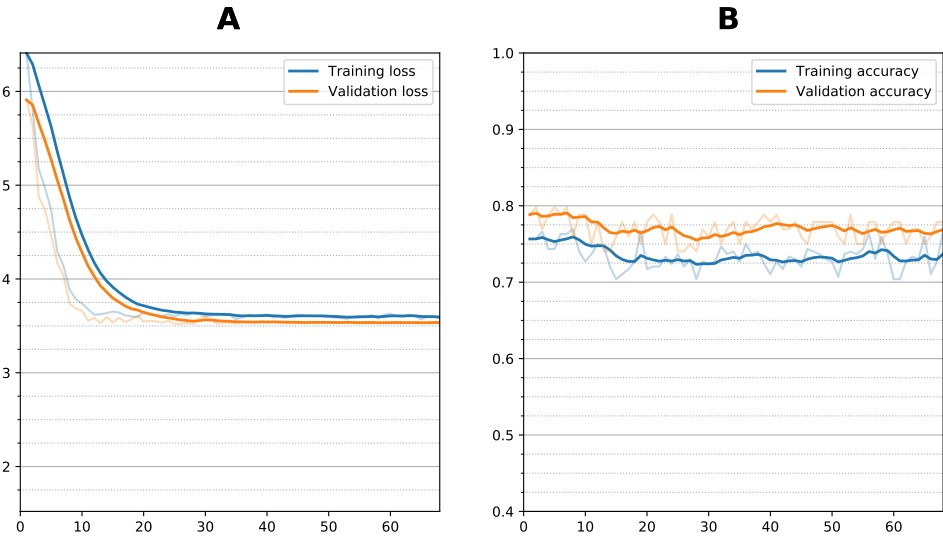

**Figure 6  Training curve during fine tuning for prediction.** (A) Loss function per epoch and (B) accuracy per epoch) during transfer learning fine tuning (sequential dual channel). Weights were initialized after training with AD vs. NC and then frozen at the convolutional layers, then additional training performed with the sMCI vs. pMCI data. There is an initial reduction in loss which stabilizes after 10 epochs, with no increase in accuracy.

even though there was some clear reduction in loss function during training. Other models showed similar characteristics. Table 3 summarizes the results of the prediction experiments, showing the outcomes of the dual and single timepoint networks using both zero-shot and fine-tuning transfer learning. The bottleneck residual convolutional style, because it performed much worse for classification, was not considered for prediction. For the single timepoint experiments, the 12-month images performed better than the initial baseline images during the classification task. The dual timepoint networks performed better than the single timepoints. For zero-shot best results were obtained from the sequential model, with the dual sequential producing a 0.795 average BA against the

Ocasio and Duong (2021), *PeerJ Comput. Sci.*, DOI 10.7717/peerj-cs.560

**Table 3  Prediction of AD at 3 years.** Results (BA and AUC mean and standard deviation) of prediction using zero-shot and fine-tuning. For single-timepoint networks, resluts are shown using both the baseline and the 1-year MRIs. For zero-shot learning, each of the four-fold classification trained weights were used as-is against each of the four validation fold sets for prediction (16 attempts) and against the prediction test set (four attempts, with best result also shown). For fine-tuning, the weights from the best test zero-shot result were used as starting weights for training against each of the 4-fold validation sets.

| Model | Convolution Style | Validation mean ± SD | | | | Test mean ± SD | | | |
|---|---|---|---|---|---|---|---|---|---|
| | | Zero-shot | | Fine-tuning | | Zero-shot | | Fine-tuning | |
| | | BA | AUC | BA | AUC | BA (best) | AUC | BA | AUC |
| Single (baseline) | Sequential | 0.728 ± 0.043 | 0.790 ± 0.043 | 0.746 ± 0.0033 | 0.805 ± 0.028 | 0.765 ± 0.021 (0.79) | 0.831 ± 0.015 | 0.754 ± 0.026 | 0.834 ± 0.015 |
| | Wide Residual | 0.699 ± 0.034 | 0.775 ± 0.034 | 0.700 ± 0.017 | 0.774 ± 0.022 | 0.706 ± 0.031 (0.79) | 0.816 ± 0.024 | 0.717 ± 0.030 | 0.816 ± 0.024 |
| Single (1 year) | Sequential | 0.750 ± 0.038 | 0.807 ± 0.039 | 0.733 ± 0.050 | 0.814 ± 0.042 | 0.774 ± 0.013 (0.79) | 0.857 ± 0.012 | 0.728 ± 0.008 | 0.836 ± 0.001 |
| | Wide Residual | 0.729 ± 0.038 | 0.799 ± 0.028 | 0.704 ± 0.037 | 0.782 ± 0.033 | 0.743 ± 0.029 (0.77) | 0.834 ± 0.020 | 0.719 ± 0.007 | 0.803 ± 0.001 |
| Dual | Sequential | 0.751 ± 0.027 | 0.808 ± 0.029 | 0.712 ± 0.027 | 0.772 ± 0.039 | 0.795 ± 0.010 (0.80) | 0.874 ± 0.009 | 0.739 ± 0.012 | 0.828 ± 0.007 |
| | Wide Residual | 0.727 ± 0.038 | 0.806 ± 0.033 | 0.719 ± 0.018 | 0.801 ± 0.032 | 0.753 ± 0.034 (0.79) | 0.842 ± 0.010 | 0.775 ± 0.003 | 0.834 ± 0.001 |

Notes.

AUC,  area under the receiver operating characteristic curve;  BA,  balanced accuracy;  SD,  standard deviation.

test set, followed by the single timepoint sequential (using the 1-year image as input) with a BA of 0.774. Transfer learning with fine tuning in all cases resulted in *lowered* accuracy as compared with the zero-shot approach. This occurred whether fine tuning was attempted by unfreezing only the fully connected layers or also the last convolutional layer. Training time for each fine-tuning experiment was approximately 15–30 min. Prediction of conversion took using a trained model (either zero-shot or fine-tuned) took two seconds or less.

To visualize the brain regions that are most relevant to ML algorithms, post-training heatmaps for the wide residual dual network were generated (Fig. 7). The highlighted structures included the lateral ventricles, periventricular white matter, and cortical surface gray matter.

## DISCUSSION

This study developed and evaluated a few sophisticated ML algorithms to predict which MCI patients would convert to AD at three years post-diagnosis using longitudinal whole-brain 3D MRI without a priori segmentation of regional structural volumes or cortical thicknesses. MRI data used for prediction were obtained at baseline and one year after baseline. The sequential convolutional approach yielded slightly better performance than the residual-based architecture, the zero-shot transfer learning approach yielded better performance than fine tuning, and the CNN using longitudinal data performed better than the CNN using a single timepoint MRI in predicting MCI conversion to AD. The best CNN model for predicting MCI conversion to AD at 3 years after diagnosis yielded a BA of 0.793.

Our predictive model used whole-brain MRIs without extraction of regional brain volumes and cortical thicknesses. We also evaluated multiple longitudinal network configurations (i.e., Siamese and non-Siamese twin networks with subtraction and concatenation as the merge function). Longitudinal images were found to be optimally processed by a twin architecture with concatenation merge. The dual timepoint network performed better regardless of whether the initial or the follow up image was used for the single timepoint. Restricting the network in a Siamese configuration where the weights of both channels are identical or using a subtraction merge function resulted in worse prediction, which suggests that the networks take full advantage of the additional information provided by the second time point data when they were allowed to train each channel with separate weights.

We employed 3D MRI instead of 2D multi-slice MRI. Previous studies have also reported MCI to AD prediction using a sequential full volume 3D architecture have obtained BAs of 0.75 (*Basaia et al., 2019*) and 0.73 (*Wen et al., 2020*), while a study using residual architecture showed a resulting BA of 0.67 (*Shmulev & Belyaev, 2018*) but did not involve longitudinal MRI data. Some studies used predetermined 3D patches uniformly sampled across the brain (*Lian et al., 2018*; *Liu et al., 2018*; *Wen et al., 2020*). A limitation of the 3D-patch approach is that a subsequent fusion of the results via some kind of ensemble or voting method is needed to obtain a subject-level prediction, and brain-wide anatomic relationships are not taken into account.
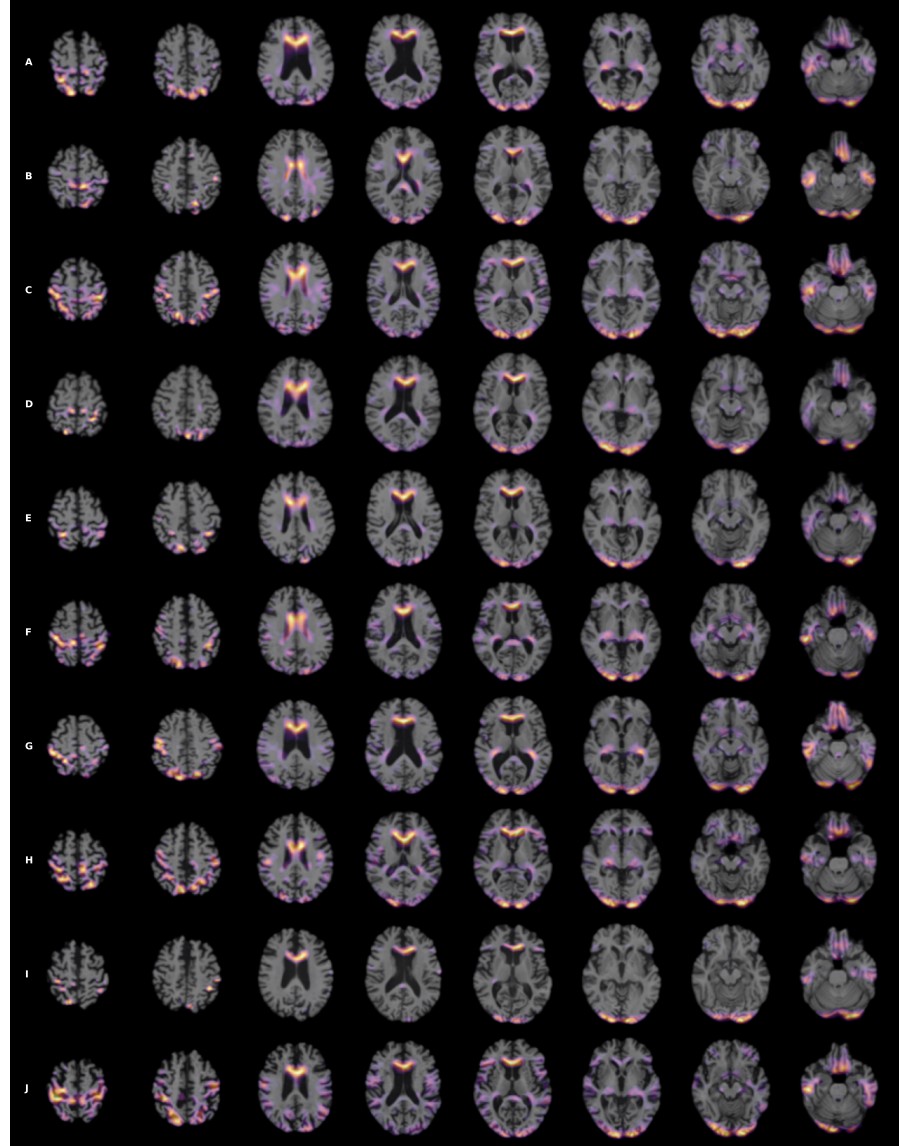

**Figure 7** **(A-J) Heatmap visualization for 10 patients.** 3D Grad-CAM heatmaps from the wide residual dual channel network used to predict conversion of MCI to AD. Heat maps were superimposed on individual patient's anatomical MRI of 10 patients. A areas in bright yellow-orange (low to high) color corresponding to voxels with the gradient based on 3D Grad-CAM algorithm at the convolutional layer around 20 pixel resolution.

There are two previous related studies that used longitudinal MRI data for prediction of MCI or AD disease progression. Bhagwat et al. employed baseline and 1-year MRIs with Siamese neural network with concatenation merge to predict a pattern of decline in patients' MMSE score, yielding an accuracy of 0.95 (*Bhagwat et al., 2018*). In contrast to our study, regional cortical thicknesses, a non-convolutional method, and clinical variables were used. The use of clinical variables could have contributed substantially to higher

accuracy. Ostertag et al. used a similar Siamese network but employed whole-brain MRI to predict decline in patients' MMSE score, yielding a validation accuracy of 0.90, but no independent evaluation on a separate test dataset was performed (*Ostertag, Beurton-Aimar & Urruty, 2019*). Moreover, these two studies differed from ours in that they mixed AD, NC, and MCI patients together, and thus their prediction accuracies are not directly comparable to those from MCI to AD conversion studies because the baseline diagnosis of NC or AD by itself is a strong predictor of neurocognitive decline.

The use of a Siamese network architecture to analyze longitudinal changes in disease progression from medical images was explored by Li et al. and specifically studied in AD brain MRIs by Bhagwat et al. and Ostertag et al. (*Bhagwat et al., 2018*; *Ostertag, Beurton-Aimar & Urruty, 2019*; *Li et al., 2020*). The idea behind Siamese networks is that both images are processed by the convolutional layers with identical parameters, with equivalent flattened sets of features for each image at the end of the convolutions. Thus, theoretically, a direct subtraction merge of the corresponding flattened features would represent a measure of the progression of the images—presumably an MCI patient whose structural MRI features have worsened in a year would be more likely to progress to AD than a patient whose features remain stable. However, a simple subtraction merge may result in loss of predictive information if there are particular features that are predictive of progression regardless of whether they have changed between baseline and 1-year. Thus, we also explore the concatenation merge. In addition, a twin (non-Siamese) network with separate parameters may provide, partly due to the additional power of doubling the number of convolutional parameters, better predictive capacity, so we also explored this architecture. Since the flattened set of post-convolution features in the twin architecture is different in each channel, as they are the result of different parameters, there is no rationale for directly subtracting them, so we only considered a concatenation merge option for the twin architecture.

Although for the initial classification task, the twin wide residual network performed best among all architectures, after the transfer learning the twin sequential network was the overall best performer. In the single channel variants, the sequential networks performed best. The bottleneck variant of the residual network performed the worst amongst all architectures. In general, the residual networks provide the benefit of reducing the vanishing gradient problem, as compared with a non-residual sequential style. The bottleneck in particular is meant to strongly prevent vanishing gradients. Since vanishing gradients did not appear significantly during training, the advantages of the residual network appeared not to materialize, and thus, overall, the sequential networks seemed best fit for 3D MRI whole-brain analysis.

## Heatmaps

Heatmaps enabled visualization of the brain regions that were most relevant to ML algorithms to predict MCI and AD conversion. The most salient structures on the heatmaps were the lateral ventricles, periventricular deep white matter as well as extensive cortical gray matter. Ventricular enlargement and atrophy are known to be associated with AD. Reduction in white-matter volume has been described in AD, including some the specific

regions that our heatmap analysis found to be on interest (*Smith et al., 2000*; *Guo et al., 2010*; *Kao et al., 2019*), including the cingulate gyrus (*Brun & Gustafson, 1976*; *Hirono et al., 1998*; *Jones et al., 2006*), the middle occipital gyrus (*Zhang & Wang, 2015*), and the putamen (*Pini et al., 2016*). Other brain regions that have shown to be associated with development of AD, such as the default node network and hippocampus, are not uniformly highlighted in the heatmaps. Our analysis approach is different from previous analysis and does not specifically identify networks, although amongst the heatmaps shown, there were components that were part of the default mode networks and hippocampus. In other words, our analysis did not specifically test whether hippocampus or default mode networks are predictive of MCI to AD conversion. It is possible that, since our MRI is based on structural changes, hippocampus and default mode networks might not have developed atrophy to be informative to prediction conversion.

## Other technical considerations

We examined three different convolutional architectures to identify the best performance prediction model. Two residual variants were compared, with the wide residual network performing better than the bottleneck variant, and the non-residual sequential network performing better than both residual types. The two residual approaches compared here were 3D modifications of ResNet (*He et al., 2016*). The bottleneck variation used pre-activation, a technique where the batch normalization and activation layers precede the convolutions. The term "bottleneck" refers to a design where each residual block includes two initial layers with narrower widths. The second residual variant examined for comparison was the wide residual network (*Zagoruyko & Komodakis, 2016*). In this approach the widths were progressively increased, with an additional dropout layer between two convolutions in each residual block. The sequential model we tested was a 3D extension of the 2D VGG model (*Simonyan & Zisserman, 2014*), with sequential blocks formed by a combination of convolutional layers followed by pooling layers.

We also examined the relative performance of two transfer learning approaches. Zero-shot technique performed better than fine tuning. Further fine-tuning with the sMCI vs pMCI data reduced the accuracy of the prediction network from that obtained via the exclusive use of AD vs NC data for *classification* task training. The lack of training power of the MCI data suggests that brain images with either AD or NC, with their more discriminant anatomic features, are more suited for training a network eventually used for detecting the more subtle distinctions between pMCI and sMCI.

We also carefully prevented data leakage by splitting the training and testing datasets at patient level, ensuring that no data from the same patient would end up in both groups (*Wen et al., 2020*). Another type of leakage we avoided occurs when data are used for training the classification are also used for prediction task. Finally, in this study the testing set results were collected only after all training was completed to prevent a third possible kind of leakage, namely where results from the test set influence the selection of hyperparameters or architecture. We also excluded patients who converted from MCI to AD before their 12-month follow up.

In several cases for both classification and prediction we observed that the BA for the testing dataset had a slightly higher mean and lower standard deviation than the corresponding results for validation. This higher variation in the validation experiments could potentially be explained by the fact that each cross-validation fold has a different validation set of images while all the testing results are obtained from the same single test set applying the different trained models. This higher variation also means that a single lower BA result in one of the validation folds could pull down the mean validation BA.

## Limitations and future directions

The increase in BA obtained by using the longitudinal MRI (0.795 vs 0.774) was modest, although both techniques represented an increase as compared to other published predictions of MCI conversion to AD. If the longitudinal MRI is otherwise available, it seems evident that the incremental improvement in predictive accuracy would justify its use. It is unclear, however, that without other reasons for performing a 1-year follow up MRI, this increase in predictive accuracy would represent a new indication from a cost-effectiveness perspective. Thus, a comprehensive cost-benefit model analysis would be useful in this area.

The study used only anatomical MRI data. Multiparametric MRI (such as diffusion-tensor imaging, task functional MRI and resting-state MRI) will be incorporated into these models in the future. Similarly, other modalities such at Positron Emission Tomography (PET) and non-imaging clinical data can also be included in the model. Further studies will need to apply this approach to other datasets to improve generalizability. Future studies should investigate MCI to AD conversion at 1, 2 and 5 years post-diagnosis.

Our model is a predictive model approach that employs machine learning based on whole-brain anatomical MRI to predict MCI to AD conversion. Future studies will need to compare different predictive models including those that predict MCI to AD conversion based on extracted volume and cortical thickness as obtained using tools such as FastSurfer (*Henschel et al., 2020*). To do so, we will first systematically explore various methods to extract volume and cortical thickness, explore various approaches (such neural networks and support vector machines) to predict MCI to ADC conversion, and use these methods to do head-to-head comparisons on the same datasets.

Deep survival analysis (*Ranganath et al., 2016*) has been applied to the prediction of conversion to AD. Future studies should also combine imaging with non-imaging data, such as neurocognitive scores (*Nagaraj & Duong, 2021*) in predictive models using ML. Nakagawa et al. used deep survival analysis to model the prediction of conversion from either MCI or NC subjects to AD using volumetric data from MRI (*Nakagawa et al., 2020*). A future extension of this analysis should investigate the use of data from the CNN models, both single-channel and longitudinal, using features extracted at the end of the convolutional layers.

## CONCLUSIONS

This is the first convolutional neural network model using longitudinal and whole-brain 3D MRIs without extracting regional brain volumes or cortical thicknesses to predict future

MCI to AD conversion. This framework set the stage for further studies of additional data time points, different image types, and non-image data to further improve prediction accuracy of MCI to AD conversion. Accurate prognosis could lead to better management of the disease, thereby improving the quality of life.

### Funding
The authors received no funding for this work.

### Competing Interests
The authors declare there are no competing interests.

### Author Contributions
- Ethan Ocasio and Tim Q. Duong conceived and designed the experiments, performed the experiments, analyzed the data, performed the computation work, prepared figures and/or tables, authored or reviewed drafts of the paper, and approved the final draft.

### Data Availability
The raw data is open and publicly available at the Alzheimer's Disease Neuroimaging Initiative (ADNI): Available at http://adni.loni.usc.edu/data-samples/access-data/. ADNI requires an application to access the data.

Patient and image IDs used for training, validation, and testing and the Python code are available in Supplementary Files.

### Supplemental Information
Supplemental information for this article can be found online at http://dx.doi.org/10.7717/peerj-cs.560#supplemental-information.

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
