# Peer review of "Deep learning prediction of mild cognitive impairment conversion to Alzheimer’s disease at 3 years after diagnosis using longitudinal and whole-brain 3D MRI"

_PeerJ Computer Science, doi:10.7717/peerj-cs.560_

## Round 0.1 · original submission · Major Revisions

I apologize for the delay in processing your article, but we have now received three expert reviews of your article; I will not re-state all the comments here, but I urge you to read and consider them carefully, including particularly comments by multiple reviewers about methods/analysis and about sharing or posting code, data and methods checklist. The reviewers also ask for more information about how your results can be interpreted in the context of existing literature.

Reviewer 1 ·

Basic reporting

Authors says "Future studies should investigate MCI to AD conversion at 1, 2 and 5 years". Recently, regarding this topic, deep survival analysis has been developed and applied to MRI data. Authors can add discussion about this.

Experimental design

no comment

Validity of the findings

Authors used 4-Fold cross validation. Does this method need to be repeated more than once to avoid subject selection bias?

Many studies revealed default mode network and hippocampus contributes development of AD and MCI. Why doesn't heatmap in this study show their contributions? Please add discussion on this if possible.

Additional comments

In abstract, what does "#D" means?

I strongly recommend authors to upload the script on web.

Reviewer 2 ·

Basic reporting

In the present study, the authors investigate the performance of several deep learning models using structural MRI for the prediction of MCI conversion to AD dementia within 3 years. Furthermore, the authors explored the added value of including a second structural MRI performed 12 months after baseline; the authors thus had to exclude MCI participants who converted to AD dementia within 1 year. This implies that their prediction model was in fact a prediction model over 2 years in MCI subjects that remained stable over 1 year. The paper is well written and easy to follow.

Experimental design

Methods are clearly explained. The authors should have followed the TRIPOD guidelines for predictive models and provide the corresponding checklist (https://www.equator-network.org/reporting-guidelines/tripod-statement/). The authors should explore how changing random partitions for testing influences their results.

Validity of the findings

Their main novel finding was that including longitudinal MRI improved the prediction of conversion from MCI to AD dementia. However, this improvement was very small (0.02 in BA) and it is unlikely to be clinically relevant as the strategy of performing two structural MRI scans over a year does not seem to be sufficiently cost-effective. Moreover, the authors have not even demonstrated that their novel model actually outperforms standard regional volumetric measures such as hippocampus volume or medial temporal thickness that can be rapidly measured with other deep learning approaches such as e.g. FastSurfer. Therefore, with the data presented here, I cannot see how this model actually represents an advance towards better prediction of AD dementia in MCI patients.

·

Basic reporting

The writing is fluid, and the text is reasonably easy to follow.
In general, there is sufficient background, and the reporting is sound.
For paper improvement, there are some remarks that need to be addressed, described below. Some details are missing, and some clarification is needed.


- Minor writing problems, for instance, "T1-weighted #D MRI" (line 27), "some used predetermination 3D patches" (line 90). Please review.

- "improve or slow the rate of decline of symptoms" (line 64). Even though the meaning could be inferred, this exact sentence is confusing. Please rewrite.

- "In contrast to traditional analysis methods such as logistic regression, ML does not require relationships between different input variables and the outcomes to be explicitly specified a priori" (lines 69-71). Actually, logistic regression is also a form of ML. Please use another example.

- "To our knowledge, there are no published studies to date on deep learning to predict MCI to AD conversion using longitudinal and whole-brain 3D MRI." (lines 100-101). In the cited paper "Convolutional Neural Networks for Classification of Alzheimer’s Disease: Overview and Reproducible Evaluation", there are some references to the combination of 3D subject-level CNN (meaning whole-brain), Longitudinal, and sMCI vs pMCI, for instance, in Tables 5 and 6. Please clarify how the proposed method is different from this paper.

- "main binary classifier metric" (line 161). Given that BA only works at a determined threshold, it would be also interesting to report AUC, as it provides a general performance of the model across a range of thresholds.

- "The neural network models [...]" (lines 173-). The network models are simply stated. I feel there is a need to add a discussion on the reasoning for choosing these models.

- "After initial analysis of networks with varying heads, global maximum pooling followed directly by a single final dense prediction layer was selected as the optimal fully connected layer architecture." (lines 194-196). Given this description, "heads" in Figures 2 and 4 might actually be misleading. Please adjust accordingly.

- "Resulting images had a resolution of 91x109x91 voxels." (lines 148-149). Considering down-sampling was performed in halves (stride=2 in Figure 3), how did authors deal with odd dimensions throughout the network? Additionally, given down-sampling was symmetrical, how did 91x109x91 voxels end up as 2x2x2 voxels or less? Please clarify.

- "L2 regularization as also added in the convolutional layers in all networks." (line 211). Please include the regularization parameter.

- "All training was performed using Tensorflow2/Keras python library, on Google Compute Platform virtual instances with Tesla V-100 GPU acceleration." (lines 211-213). It would be interesting to include some information on training and inference times.

- "Discussion" (lines 259-); "We examined three different convolutional architectures [...]" (lines 314-). I feel that, at some point in the discussion, an analysis about the performances from the different networks is missing, for instance, why an architecture would be better than the others. Please complement.

Experimental design

This research is relevant and meaningful.
For paper improvement, there are some points in the text that need better descriptions and clarification. For a rigorous investigation, a few additional experiments are necessary.


- "(3) a twin network (identical channels with weights independently optimized)." (lines 180-181). The siamese networks were experimented with two merging options. It seems that the twin network was experimented with only one merging option. Which one? Why? Please clarify. For completeness, it would be interesting to experiment with both options using the twin network as well.

- "Data were split into 75% and 25% for training and testing, respectively, with 4-fold cross validation." (lines 34-35)
- "Data for training and testing were randomly split up as 75% and 25%, with the training set composed of 415 patients (249 sMCI and 166 pMCI), and the testing set composed of 139 patients (84 sMCI and 55 pMCI). Training employed 4-fold cross validation." (lines 129-132)
- "Data for training and testing were randomly split up as 70% and 30%. The training set consisted of 387 patients (160 AD and 227 NC). Training employed 4-fold cross-validation." (lines 136-138)
- "Images were split into testing, validation, and testing sets at the patient level in order to avoid data leakage. [...]" (lines 154-159)
- "We also carefully prevented data leakage [...]" (lines 336-)
I appreciate that the authors were concerned with data leakage, as this is very important. However, the text is not completely clear, and, more importantly, I'm afraid that the provided code does not represent what is described in the text.
In each data split, it seems that cross-validation was performed on the "training" set, which makes it a combined training and validation set. If this is correct, a possible suggestion would be in the lines of: "Data were split into 75% and 25% for training/validation and testing, respectively. Then, we optimized the networks using a 4-fold cross-validation on the training/validation set, resulting in 56.25% for training and 18.75% for validation from the complete data in each fold split. In the end, we had four trained models for each experiment configuration, and reported the mean and SD BA."
Additionally, in file data_split.py, there is a single call to train_test_split when splitting training and validations sets. It seems that there is no cross-validation. Please double check.

- "During training, data augmentation was performed on the training set by rotating each MRI by up to 5% in any direction" (lines 149-150). Given that registration includes rotation, I wonder how much this rotation augmentation might actually improve (or even deteriorate) results. For completeness, I suggest the authors perform an additional experimentation without this rotation augmentation.

- "For the single timepoint experiments, we used the 12-month images because they performed better than the initial baseline images during the classification task." (lines 245-247)
- "The dual timepoint network performed better regardless of whether the initial or the follow up image was used for the single timepoint." (lines 274-275)
These are very important statements. If the 12-month images were used in the classification task, the results shouldn't change, as both baseline and 12-month images represent either AD or CN. However, if the 12-month images were used in the prediction task, then it changes the interpretation of the experiments and results, as it would mean a prediction at 2 years, instead of 3 years.
Nevertheless, using the follow up image for the single timepoint is an interesting experiment and should be reported throughout the paper, in Methods, Results, and Discussion.
Note that, in Figure 2, there are some differences between the architectures. First, as previously stated, there is the difference between baseline and follow up images, which should be compared in the single timepoint experiment. Secondly, the siamese architecture has twice the amount of input data. Finally, the twin architecture has twice the amount of input data and nearly twice the amount of parameters. I see this as an ablation study going from the "baseline" model to the "final/best" model.
For completeness, please report both single timepoint experiments.

Validity of the findings

It is not possible to completely assess the validity of the findings at this moment due to questions raised in "2. Experimental design".

---

## Round 0.2 · Minor Revisions

You will see that the reviewer feels you have adequately addressed most of the concerns with your original submission, but suggests a number of fairly straightforward remaining changes, and I concur with most of them. Please read and see if you agree that these are well-founded and, if so, address them in a revision. If there are any reviewer comments with which you strongly disagree, please explain in your rebuttal letter. Assuming that you are able to make all/most of the requested changes, and that you have justifiable reasons for any that you decide not to implement, I would likely issue a determination of Accept without need for re-review at that point. However, this is not guaranteed.

·

Basic reporting

I appreciate the authors incorporating the observations made in the reviews.
There are only a few questions/remarks left.

"slow the rate of cognitive decline" (line 22)
"to improve symptoms, or at least slow their rate of decline" (line 64)
Please note that English is not my native language, so take this with a grain of salt.
The first sentence makes sense to me, while the second one does not.
In the first one, you are trying to slow the rate of cognitive decline. In the second one, you are trying to slow the rate of symptoms decline.
In my understanding, a symptom is something bad, such as bad memory. When a symptom declines, I infer that the patient is getting better. If you slow this rate, then the patient will no longer get better. That's why I suggested rewriting this sentence.

Table 2 and Table 3
It is not common to see test performance better than validation performance. Do the authors have an opinion on why this happened? It might be interesting to include some discussion about this.

"The neural network models [...]" (lines 189-)
I appreciate the authors adding a discussion about siamese networks and merging options.
I feel that the same discussion could be included regarding the types of convolutional blocks ("sequential, residual with bottleneck, and wide residual").

"After initial analysis of networks with varying heads, global maximum pooling followed directly by a single final dense prediction layer was selected as the optimal fully connected layer architecture." (lines 224-226)
I apologize for not being clear.
My understanding from this sentence is that, after the conv or merge blocks, there is a global maximum pooling layer, and a single final dense prediction layer (with 1 hidden unit, aka, neuron). There is a mismatch between this understanding and what is represented by "head" in Figures 2 and 4.
Figure 2 shows three FC layers. Figure 4A shows two FC layers. I believe that they should match.

"When a convolutional layer processes an input whose size is odd-numbered in any of its dimensions (2n-1 for any integer n) the resulting output of a stride 2 convolution will be of size n for the corresponding dimension." (lines 214-216)
I appreciate the clarification. I believe that this behavior really depends on the implementation. So I understand that the implementation used here adds a zero-padding before the convolution, so no information is lost.

Experimental design

"Using either the single timepoint with the 1-year image or both timepoints together longitudinally represents, for patients who have had MCI for one year and not yet progressed to AD, a prediction of whether they will eventually convert to AD within two more years." (lines 203-206)
Given this, maybe the authors should adjust the title (and respective text in the paper) to 2 years.

Validity of the findings

Findings are sound and grounded.

---

## Round 0.3 · accepted · Accept

I find that you have successfully addressed all remaining reviewer concerns, and I hope that you will agree that the manuscript is stronger as a result. Your article should make a nice contribution to the literature.